# FM-TS: FLOW MATCHING FOR TIME SERIES GENERATION

## ABSTRACT

Time series generation has emerged as an essential tool for analyzing temporal data across numerous fields. While diffusion models have recently gained significant attention in generating high-quality time series, they tend to be computationally demanding and reliant on complex stochastic processes. To address these limitations, we introduce FM-TS, a rectified Flow Matching-based framework for Time Series generation, which simplifies the time series generation process by directly optimizing continuous trajectories. This approach avoids the need for iterative sampling or complex noise schedules typically required in diffusion-based models. FM-TS is more efficient in terms of training and inference. Moreover, FM-TS is highly adaptive, supporting both conditional and unconditional time series generation. Notably, through our novel inference design, the model trained in an unconditional setting can seamlessly generalize to conditional tasks without the need for retraining. Extensive benchmarking across both settings demonstrates that FM-TS consistently delivers superior performance compared to existing approaches while being more efficient in terms of training and inference. For instance, in terms of discriminative score, FM-TS achieves 0.005, 0.019, 0.011, 0.005, 0.053, and 0.106 on the Sines, Stocks, ETTh, MuJoCo, Energy, and fMRI unconditional time series datasets, respectively, significantly outperforming the second-best method which achieves 0.006, 0.067, 0.061, 0.008, 0.122, and 0.167 on the same datasets. We have achieved superior performance in solar forecasting and MuJoCo imputation tasks, significantly enhanced by our innovative $t$ power sampling method.

## 1 INTRODUCTION

Time series data is fundamental to modern data analysis, serving as a cornerstone in diverse domains such as finance, healthcare, energy management, and environmental studies (Lim and Zohren, 2021; Ye et al., 2024; Dama and Sinoquet, 2021; Liang et al., 2024). However, acquiring high-quality time series data often presents significant challenges, including stringent privacy regulations, prohibitive data collection costs, and data scarcity in certain scenarios. These challenges highlight the potential benefits of synthetic time series data, which can provide a cost-effective solution for data scarcity, overcome privacy concerns, and offer flexibility in generating diverse scenarios representing a wide range of possible patterns and trends. To obtain high-quality synthetic data, there is a pressing need for advanced time series generation techniques that can produce realistic and diverse patterns, accurately reflecting real-world complexities and variations.

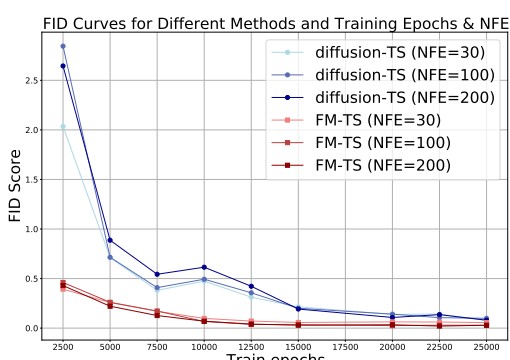

Figure 1: Comparison of FM-TS and diffusion-TS in terms of efficiency on Energy dataset under varying training epochs and number of forward evaluation steps.

Recent years have witnessed significant advancements in time series generation, ranging from VAE-based approaches (Desai et al., 2021; Xu et al., 2020) to diffusion models (Kong et al., 2021; Tashiro et al., 2021), demonstrate remarkable capabilities in capturing complex temporal dynamics.

While these studies have paved new paths for time series modeling (Coletta et al., 2023; Yoon et al., 2019a), important challenges remain in theoretical foundations and computational efficiency. Diffusion models (Ho et al., 2020; Song et al., 2020a;b) are then utilized for time series generation, yield exceptional generative quality. They offer several advantages, including their ability to capture long-range dependencies and generate diverse, high-quality samples. However, diffusion models suffer from slow generation speeds and high computational cost due to the requirement of many steps to infer (see figure 1 and (Nichol and Dhariwal, 2021)). Moreover, diffusion models struggle to preserve the long-term dependencies and intricate patterns inherent in time series data (Rasul et al., 2021).

Recently, rectified flow matching (Liu et al., 2022) has emerged as a promising generative modeling approach, because of its efficiency and capacity for scalability (Esser et al., 2024a). Rectified flow matching optimizes neural ordinary differential equation (ODE) to transport between distributions along approximately straight paths, solving a nonlinear least squares problem. This approach offers more efficient sampling than diffusion models through approximately straight paths, while providing a unified framework for generative modeling and domain transfer with theoretical guarantees on transport costs (Liu et al., 2022).

In contrast to diffusion models, rectified flow matching directly maps the latent space to the data space, whereas diffusion models must learn to denoise data based on a scheduled noise-adding process. In addition, rectified flow matching requires only a single forward pass for sampling (Liu et al., 2022), significantly enhancing both efficiency and performance. Rectified flow matching has shown superior performance in various tasks, including image generation (Kim et al., 2024; Mehta et al., 2024; Kuaishou Technology, 2024). However, it has not yet been applied to time series generation, primarily due to the unique characteristics of time series data, such as temporal dependencies and potential seasonality.

To address these challenges, we introduce FM-TS, a flow matching based framework for time series generation. Our method not only inherits the efficiency of rectified flow matching but can also generalize in both unconditional and conditional settings. The main contributions of this work are:

- FM-TS consistently outperforms existing state-of-the-art methods across a variety of time series generation datasets with notable efficiency (see Figure 1). To the best of our knowledge, this work is the first to utilize rectified flow matching to time series generation.

- For conditional time series generation, we also introduce a simple yet powerful sampling technique: $t$ power sampling, a simple timestep shifting method (used in generation), which can boot performance of conditional generation quite a lot.

- With our novel inference design, the model trained in an unconditional setting can seamlessly generalize to conditional tasks without requiring retraining and redundant gradient-based optimization steps like (Yuan and Qiao, 2024).

The experiments on various tasks demonstrate that the proposed framework can significantly boost performance through rectified flow matching. We achieve most state-of-the-art, e.g., FM-TS can achieve context fid (lower is better) with 0.019, 0.011 on stocks, ETTh unconditional generation datasets while previous best result is 0.067, 0.061. On solar forecasting tasks, our method achieves an MSE of 213, outperforming the previous best result of 375 (Yuan and Qiao, 2024) by 43.2%.

## 2 RELATED WORK

### 2.1 TIME SERIES GENERATION

Generating realistic time series data has attracted significant attention in recent years, driven by the need for high-quality synthetic data in various domains such as finance, healthcare, and energy management (Lim and Zohren, 2021). Unconditional time series generation (Nikitin et al., 2023) is to generate time series data without specific constraints to mimic statistical properties and patterns of real data. Conditional time series generation is to Generate time series data based on specific conditions or constraints, like forecasting (Alcaraz and Strodthoff, 2022a) and imputation (Tashiro et al., 2021). Early time series generation approaches primarily utilized Generative Adversarial Networks (GANs) (Goodfellow et al., 2014). Notable works in this category include TimeGAN (Yoon et al., 2019a),

which incorporates an embedding network and supervised loss to capture temporal dynamics, and RCGAN (Esteban et al., 2017), which uses a recurrent neural network architecture conditioned on auxiliary information for medical time series generation. Both TimeGAN and RCGAN are capable of conditional generation, with RCGAN specifically designed for conditional tasks, while TimeGAN can be adapted for both conditional and unconditional generation.

## 2.2 DIFFUSION MODELS FOR TIME SERIES

Recently, diffusion models, particularly Denoising Diffusion Probabilistic Models (DDPMs) (Ho et al., 2020), have emerged as a powerful paradigm for generative modeling across various domains. Diffusion models offer better perceptual quality compared to GANs, avoiding optimization issues in adversarial training. In the context of time series, diffusion models have shown promising results in tasks such as audio synthesis (Kong et al., 2020), time series imputation (Tashiro et al., 2021), and forecasting (Rasul et al., 2021). (Rasul et al., 2021) proposed TimeGrad, a conditional diffusion model that predicts in an autoregressive manner, guided by the hidden state of a recurrent neural network. Tashiro et al. (2021) and Alcaraz and Strodthoff (2022a) adapt diffusion models for time series imputation using self-supervised masking strategies. Shen and Kwok (2023) introduced TimeDiff, a non-autoregressive diffusion model that addresses boundary disharmony issues in time series generation. For unconditional time series generation, Lim et al. (2023) employed recurrent neural networks as the backbone for generating regular 24-time-step series using Score-based Generative Models (SGMs). Kollovieh et al. (2024) proposed a self-guiding strategy for univariate time series generation and forecasting based on structured state space models. However, these methods suffer from slow generation speeds, high computational costs, and a complex sampling schedule.

## 2.3 FLOW MATCHING FOR GENERATION

Rectified flow matching (Liu et al., 2022) is a simple ODE method for high-quality image generation and domain transfer with minimal steps, differing from diffusion models by avoiding noise and emphasizing deterministic paths. Compared to diffusion methods, it has two main advantages, stability of training and effectiveness of inference. Rectified flow matching has shown remarkable results in video generation (Kuaishou Technology, 2024), image generation stable diffusion 3 (Esser et al., 2024b) and flux (bla, 2024), point cloud generation (Wu et al., 2023) (Kim et al., 2024), protein design (Campbell et al., 2024; Jing et al., 2024), human motion generation (Hu et al., 2023), TTS (Mehta et al., 2024; Guan et al., 2024; Guo et al., 2024). Despite the great success and effectiveness of rectified flow matching, flow matching has not yet been applied to time series generation. Witnessing the great potential of flow matching for this task, that motivates to propose FM-TS for time series generation on both unconditional and conditional settings.

## 3 METHOD

In this section, we present FM-TS, our novel framework for time series generation based on rectified flow matching. We begin by introducing the problem setting, then providing an overview of the FM-TS framework, followed by the inference pipeline of FM-TS for unconditional and conditional time series generation, respectively.

## 3.1 PROBLEM STATEMENT

**Unconditional Time Series Generation** Unconditional time series generation focuses on producing sequential data without any conditions, where the model learns underlying temporal patterns from a training set and generates new sequences that follow a similar distribution. Formally, the problem is defined as follows:

Let $X_{1:\ell} = (x_1, \ldots, x_\ell) \in \mathbb{R}^{\ell \times d}$ denote a time series covering $\ell$ time steps, where $d$ is the dimension of observed signals.

Input:   $Z_0 \sim \pi_0$;   where $Z_0 \in \mathbb{R}^{\ell \times d}$ and $\pi_0$ is $\mathcal{N}(0, I)$ .

Output:   $\hat{X}_{1:\ell} = G(Z_0) \in \mathbb{R}^{\ell \times d}$;   where $G$ transforms noise $Z_0$ into the target distribution.

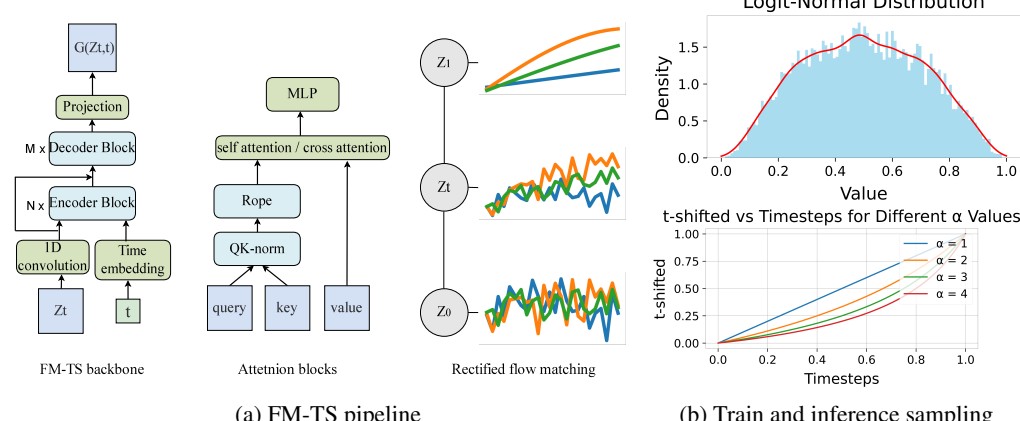

(a) FM-TS pipeline

(b) Train and inference sampling

Figure 2: **Overview of FM-TS.** (a) FM-TS pipeline. It use $G$ as the model, which takes $Z_t$ and $t$ as input to generate outputs $G(Z_t, t)$ (see Eq. 3). The attention blocks in encoder/decoder blocks of $G$ is specifically designed shown in the middle. The overall idea of learning rectified flow from $Z_0$ to $Z_1$ is illustrated in the right panel, where $Z_t$ is a linear interpolation of $Z_0$ and $Z_1$ at timestep $t$. (b) The sampling strategy of training and inference. Logit-normal sampling can help the model to focus on learning the hardest part (when t is around 0.5). The t-shifting sampling in inference can generate results with better quality.

Common generative models used for $G$ include GANs (Yoon et al., 2019b), VAEs (Desai et al., 2021), and diffusion models (Tashiro et al., 2021; Yuan and Qiao, 2024), which are capable of capturing complex temporal dependencies. During training process, $G$ is optimized via different strategies to minimize the difference between output $\hat{X}_{1:\ell}$ and target $X_{1:\ell}$.

**Conditional Time Series Generation** Conditional time series generation produces sequences based on partially known data, utilizing the prior information as context. The generated sequence contains both the observed and predicted segments. Formally:

$$\text{Input:} \quad Z_0 \sim \pi_0, \quad y \in \mathbb{R}^{m \times d}; \quad \text{where } Z_0 \in \mathbb{R}^{\ell \times d}, \quad \pi_0 \text{ is } \mathcal{N}(0, I),$$

$$y \in \mathbb{R}^{m \times d} \text{ is the observed time series with length } m \text{ (where } m < \ell).$$

$$\text{Output:} \quad \hat{X}_{1:\ell} = G(Z_0, y) \in \mathbb{R}^{\ell \times d};$$

$$\text{where } G \text{ transforms noise } Z_0 \text{ into the target distribution conditioned on } y.$$

Here $G$ includes same generative models as unconditional models above.

For conditional time series generation, this can be further categorized into 2 main directions:
1) **Forecasting**: $G$ is trained as forecasting functions that maps past observation to future predictions given $y = (x_1, x_2, ..., x_m)$.
2) **Imputation**: The model $G$ is trained to fill in missing values at unobserved timesteps, given that $y$ is derived from $m$ observed timesteps within the range of 1 to $\ell$.

The difference between forecasting and imputation is the position of known values. The mask $M \in \mathbb{R}^{\ell \times d}$ indicating the known/missing values which will be used in Algorithm 1.

## 3.2 RECTIFIED FLOW MATCHING FOR TIME SERIES GENERATION.

In FM-TS, we propose to learn rectified flow as the model $G$ for time series generation. Rectified flow (Liu et al., 2022) is a method of learning ordinary differential equation (ODE) models to transport between two empirical distributions $\pi_0$ and $\pi_1$. In our setting, $\pi_0$ is $\mathcal{N}(0, I)$, and $\pi_1$ is the target distribution, where $X_{1:\ell} \sim \pi_1$. Thus, the problem can be reformulated as: given empirical observations of two distributions $Z_0 \sim \pi_0$ and $Z_1 \sim \pi_1$, find a transport map $G: \mathbb{R}^{\ell \times d} \to \mathbb{R}^{\ell \times d}$ that can map distribution $\pi_0$ to $\pi_1$. $G$ is designed to find the transport map between two distributions instead of pairwise mapping. After successful learning of $G$, we expect that $Z_1 := G(Z_0) \sim \pi_1$ when input $Z_0 \sim \pi_0$.

---

**Algorithm 1** Inference of FM-TS for conditional generation

---

**Input:**
    $\mathbf{y} \in \mathbb{R}^{l \times d}$: target time series
    $\mathbf{M} \in \mathbb{R}^{l \times d}$: observation mask, where 1,0 indicates observed and missing values, respectively.
    N: number of forward evaluations
    $G$: trained flow matching model
**Output:**
    $\hat{\mathbf{Z}}$: generated time series with condition observations.
1: $\hat{\mathbf{Z}} \sim \mathcal{N}(0, \mathbf{I}), \mathbf{Z}_0 \sim \mathcal{N}(0, \mathbf{I})$                                         ▷ Initialize $\hat{\mathbf{Z}}, \mathbf{Z}_0$
2: **for** $i \leftarrow 0$ **to** N $- 1$ **do**
3:      $t_i \leftarrow i/$N
4:      $t_i \leftarrow t_i^k$                                               ▷ $t_i$ to the power of $k$
5:      $\mathbf{Z}_0 \sim \mathcal{N}(0, \mathbf{I})$                              ▷ Reinitialize noise at each step
6:      $\mathbf{Z}_{t_i} \leftarrow t_i \hat{\mathbf{Z}} + (1 - t_i)\mathbf{Z}_0$
7:      $\mathbf{Z}_{t_i}[\mathbf{M}] \leftarrow t_i \mathbf{y}[\mathbf{M}] + (1 - t_i)\mathbf{Z}_0[\mathbf{M}]$       ▷ Replace with observed series
8:      $\mathbf{v} \leftarrow G(\mathbf{Z}_{t_i}, t_i)$                                  ▷ Flow matching step
9:      $\hat{\mathbf{Z}} \leftarrow \mathbf{Z}_{t_i} + (1 - t_i)\mathbf{v}$                           ▷ One Euler step
10: **end for**
11: **return** $\hat{\mathbf{Z}}$

---

Given the empirical observations of two distributions $Z_0 \sim \pi_0$ and $Z_1 \sim \pi_1$, the rectified flow induced from $(Z_0, Z_1)$ is an ODE on time $t \in [0, 1]$,

$$\frac{dZ_t}{dt} = v(Z_t, t), where \quad t \in [0, 1], Z_t \in \mathbb{R}^{\ell \times d} \tag{1}$$

where the drift force $v: \mathbb{R}^{\ell \times d} \to \mathbb{R}^{\ell \times d}$ is set to drive the flow to follow the direction $(Z_1 - Z_0)$ of the linear path between $Z_0$ to $Z_1$ as much as possible.

This can be achieved by solving a least squares regression problem:

$$\min_v \int_0^1 \mathbb{E}\left[\|Z_1 - Z_0 - v(Z_t, t)\|^2\right] dt \tag{2}$$

where $Z_t$ is a linear interpolation between $Z_0$ and $Z_1$: $Z_t = t \cdot Z_1 + (1 - t) \cdot Z_0$, where $v$ is expected to learn with the neural network model $G$.

Therefore, the model $G$ can be optimized by predicting the direction vector between $Z_1 - Z_0$ via the following loss function

$$\mathcal{L} = \mathbb{E}_{t \sim \text{Logit-Normal}}[\|(Z_1 - Z_0) - G(Z_t, t)\|^2] \tag{3}$$

where $G$ is the model used in FM-TS to learn the drift force $v$. For each sample, $t$ is randomly drawn from a Logit-Normal distribution (Esser et al., 2024b), while $Z_1$ is sampled from the target time series distribution $\pi_1$, and $Z_0$ is sampled from the standard normal distribution $\pi_0$.

The overview framework is demonstrated in Fig. 2. Here the unconditional time series generation model $G$ can be directly trained via loss in Eq. 3 by taking $Z_t$ and $t$ as input to predict the drift force $v$ between $Z_0$ and $Z_1$. Then the trained unconditional model can be directly used for conditional generation without the need for task-specific training of a conditional generation model.

### 3.3 SAMPLING PROCESS FOR INFERENCE

To generate new time series, we use a sampling process that follows the shifting of timestep schedules approach (Esser et al., 2024b). Starting from $Z_0 \sim \mathcal{N}(0, 1)$, we iteratively refine it using:

$$Z_{(i+1)/N} = Z_{i/N} + (t_{i+1}^{\text{shifted}} - t_i^{\text{shifted}}) \cdot G(Z_{i/N}, t_i^{\text{shifted}}) \tag{4}$$

where $N$ is the total number of iterations, and $i$ is iteratively updated from 0 to $N - 1$, $t_i^{\text{shifted}}$ is predefined time step at iteration $i$ (see Eq. 5), $G$ is the trained model.

Table 1: Unconditional time series Generation Benchmark with 24-length

| Metric | Methods | Sines | Stocks | ETTh | MuJoCo | Energy | fMRI |
|---|---|---|---|---|---|---|---|
| Discriminative Score (Lower is Better) | **FM-TS** | **0.005**±**.005** | **0.019**±**.013** | **0.011**±**.015** | **0.005**±**.005** | **0.053**±**.010** | **0.106**±**.018** |
| | Diffusion-TS | 0.006±.007 | 0.067±.015 | 0.061±.009 | 0.008±.002 | 0.122±.003 | 0.167±.023 |
| | TimeGAN | 0.011±.008 | 0.102±.021 | 0.114±.055 | 0.238±.068 | 0.236±.012 | 0.484±.042 |
| | TimeVAE | 0.041±.044 | 0.145±.120 | 0.209±.058 | 0.230±.102 | 0.499±.000 | 0.476±.044 |
| | Diffwave | 0.017±.008 | 0.232±.061 | 0.190±.008 | 0.203±.096 | 0.493±.004 | 0.402±.029 |
| | DiffTime | 0.013±.006 | 0.097±.016 | 0.100±.007 | 0.154±.045 | 0.445±.004 | 0.245±.051 |
| | Cot-GAN | 0.254±.137 | 0.230±.016 | 0.325±.099 | 0.426±.022 | 0.498±.002 | 0.492±.018 |
| Predictive Score (Lower is Better) | **FM-TS** | **0.092**±**.000** | **0.036**±**.000** | **0.118**±**.005** | 0.008±.001 | **0.250**±**.000** | **0.099**±**.000** |
| | Diffusion-TS | 0.093±.000 | 0.036±.000 | 0.119±.002 | **0.007**±**.000** | **0.250**±**.000** | **0.099**±**.000** |
| | TimeGAN | 0.093±.019 | 0.038±.001 | 0.124±.001 | 0.025±.003 | 0.273±.004 | 0.126±.002 |
| | TimeVAE | 0.093±.000 | 0.039±.000 | 0.126±.004 | 0.012±.002 | 0.292±.000 | 0.113±.003 |
| | Diffwave | 0.093±.000 | 0.047±.000 | 0.130±.001 | 0.013±.000 | 0.251±.000 | 0.101±.000 |
| | DiffTime | 0.093±.000 | 0.038±.001 | 0.121±.004 | 0.010±.001 | 0.252±.000 | 0.100±.000 |
| | Cot-GAN | 0.100±.000 | 0.047±.001 | 0.129±.000 | 0.068±.009 | 0.259±.000 | 0.185±.003 |
| | Original | 0.094±.001 | 0.036±.001 | 0.121±.005 | 0.007±.001 | 0.250±.003 | 0.090±.001 |
| Context-FID Score (Lower is Better) | **FM-TS** | **0.002**±**.000** | **0.015**±**.003** | **0.024**±**.001** | **0.009**±**.000** | **0.031**±**.004** | 0.128±.009 |
| | Diffusion-TS | 0.006±.000 | 0.147±.025 | 0.116±.010 | 0.013±.001 | 0.089±.024 | **0.105**±**.006** |
| | TimeGAN | 0.101±.014 | 0.103±.013 | 0.300±.013 | 0.563±.052 | 0.767±.103 | 1.292±.218 |
| | TimeVAE | 0.307±.060 | 0.215±.035 | 0.805±.186 | 0.251±.015 | 1.631±.142 | 14.449±.969 |
| | Diffwave | 0.014±.002 | 0.232±.032 | 0.873±.061 | 0.393±.041 | 1.031±.131 | 0.244±.018 |
| | DiffTime | 0.006±.001 | 0.236±.074 | 0.299±.044 | 0.188±.028 | 0.279±.045 | 0.340±.015 |
| | Cot-GAN | 1.337±.068 | 0.408±.086 | 0.980±.071 | 1.094±.079 | 1.039±.028 | 7.813±.550 |
| Correlational Score (Lower is Better) | **FM-TS** | 0.015±.006 | 0.012±.011 | **0.022**±**.010** | **0.183**±**.051** | **0.650**±**.201** | **0.938**±**.039** |
| | Diffusion-TS | **0.015**±**.004** | **0.004**±**.001** | 0.049±.008 | 0.193±.027 | 0.856±.147 | 1.411±.042 |
| | TimeGAN | 0.045±.010 | 0.063±.005 | 0.210±.006 | 0.886±.039 | 4.010±.104 | 23.502±.039 |
| | TimeVAE | 0.131±.010 | 0.095±.008 | 0.111±.020 | 0.388±.041 | 1.688±.226 | 17.296±.526 |
| | Diffwave | 0.022±.005 | 0.030±.020 | 0.175±.006 | 0.579±.018 | 5.001±.154 | 3.927±.049 |
| | DiffTime | 0.017±.004 | 0.006±.002 | 0.067±.005 | 0.218±.031 | 1.158±.095 | 1.501±.048 |
| | Cot-GAN | 0.049±.010 | 0.087±.004 | 0.249±.009 | 1.042±.007 | 3.164±.061 | 26.824±.449 |

The time steps $t_i^{\text{shifted}}$ is generated following stable diffusion 3 (Esser et al., 2024b)-like time shifting sampling schedule. The time shifting aims to improve the quality of high-resolution image synthesis by ensuring that the model applies the appropriate amount of noise at each timestep, which is also beneficial for time series generation. The shifting schedule is shown as Figure 2b:

$$t_i^{\text{shifted}} = 1 - \frac{\alpha \cdot t_i}{1 + (\alpha - 1) \cdot t_i} \tag{5}$$

where $t_i = i/N$ with $N$ total timesteps, and $\alpha$ is a hyperparameter. For reference, the visualization of the relationship between $t_i^{\text{shifted}}$ and $t_i$ under different $\alpha$ is shown as Fig. 2b. The larger $\alpha$ is, the more shifting scale is.

For conditional generation, a slightly different inference pipeline of FM-TS is illustrated in Algorithm 1, with the following major design changes relative to unconditional generation. ❶ $t$ **power sampling with** $k$: We find that when $k < 1$, the sampling part can focus on the later sampling steps, which can be quite useful for conditional generation. (Algorithm 1 line 4). ❷ **Add noise at each step**: The algorithm adds the noise at each step(Algorithm 1 line 5). ❸ **One Euler Step Generation**: The algorithm uses one Euler step to generate $\hat{\mathbf{Z}}$ from $Z_0$ (Algorithm 1 line 9). With the above design, FM-TS effectively combines the strengths of flow matching with conditional information, enabling guided generation of time series data.

# 4 EXPERIMENTS

## 4.1 DATASETS

Our evaluation employs six diverse datasets: The 3 real-world datasets include Stocks [1] for measuring daily stock price data, ETTh [2] (Zhou et al., 2021) for interval electricity transformer data, and Energy [3] for UCI appliance energy prediction. The 3 simulation datasets include fMRI [4] for simulated

---

[1] https://finance.yahoo.com/quote/GOOG/history?p=GOOG

[2] https://github.com/zhouhaoyi/ETDataset

[3] https://archive.ics.uci.edu/ml/datasets/Appliances+energy+prediction

[4] https://www.fmrib.ox.ac.uk/datasets/netsim/

Table 2: Benchmark of Unconditional Long-term Time Series Generation

| | Dataset | Length | FM-TS | Diffusion-TS | TimeGAN | TimeVAE | Diffwave | DiffTime | Cot-GAN |
|---|---|---|---|---|---|---|---|---|---|
| ETTh | Discriminative (Lower Better) | 64 | **0.010**±**.004** | 0.106±.048 | 0.227±.078 | 0.171±.142 | 0.254±.074 | 0.150±.003 | 0.296±.348 |
| | | 128 | **0.040**±**.012** | 0.144±.060 | 0.188±.074 | 0.154±.087 | 0.274±.047 | 0.176±.015 | 0.451±.080 |
| | | 256 | 0.081±.022 | **0.060**±**.030** | 0.444±.056 | 0.178±.076 | 0.304±.068 | 0.243±.005 | 0.461±.010 |
| | Predictive (Lower Better) | 64 | **0.115**±**.005** | 0.116±.000 | 0.132±.008 | 0.118±.004 | 0.133±.008 | 0.118±.004 | 0.135±.003 |
| | | 128 | **0.104**±**.013** | 0.110±.003 | 0.153±.014 | 0.113±.005 | 0.129±.003 | 0.120±.008 | 0.126±.001 |
| | | 256 | **0.107**±**.005** | 0.109±.013 | 0.220±.008 | 0.110±.027 | 0.132±.001 | 0.118±.003 | 0.129±.000 |
| | Context-FID (Lower Better) | 64 | **0.039**±**.003** | 0.631±.058 | 1.130±.102 | 0.827±.146 | 1.543±.153 | 1.279±.083 | 3.008±.277 |
| | | 128 | **0.128**±**.007** | 0.787±.062 | 1.553±.169 | 1.062±.134 | 2.354±.170 | 2.554±.318 | 2.639±.427 |
| | | 256 | **0.302**±**.018** | 0.423±.038 | 5.872±.208 | 0.826±.093 | 2.899±.289 | 3.524±.830 | 4.075±.894 |
| | Correlational (Lower Better) | 64 | **0.027**±**.015** | 0.082±.005 | 0.483±.019 | 0.067±.006 | 0.186±.008 | 0.094±.010 | 0.271±.007 |
| | | 128 | **0.030**±**.011** | 0.088±.005 | 0.188±.006 | 0.054±.007 | 0.203±.006 | 0.113±.012 | 0.176±.006 |
| | | 256 | **0.025**±**.008** | 0.064±.007 | 0.522±.013 | 0.046±.007 | 0.199±.003 | 0.135±.006 | 0.222±.010 |
| Energy | Discriminative (Lower Better) | 64 | 0.131±.046 | **0.078**±**.021** | 0.498±.001 | 0.499±.000 | 0.497±.004 | 0.328±.031 | 0.499±.001 |
| | | 128 | 0.301±.013 | **0.143**±**.075** | 0.499±.001 | 0.499±.000 | 0.499±.001 | 0.396±.024 | 0.499±.001 |
| | | 256 | 0.404±.070 | **0.290**±**.123** | 0.499±.000 | 0.499±.000 | 0.499±.000 | 0.437±.095 | 0.498±.004 |
| | Predictive (Lower Better) | 64 | 0.250±.009 | **0.249**±**.000** | 0.291±.003 | 0.302±.001 | 0.252±.001 | 0.252±.000 | 0.262±.002 |
| | | 128 | 0.249±.001 | **0.247**±**.001** | 0.303±.002 | 0.318±.000 | 0.252±.000 | 0.251±.000 | 0.269±.002 |
| | | 256 | 0.247±.001 | **0.245**±**.001** | 0.351±.004 | 0.353±.003 | 0.251±.000 | 0.251±.000 | 0.275±.004 |
| | Context-FID (Lower Better) | 64 | **0.058**±**.010** | 0.135±.017 | 1.230±.070 | 2.662±.087 | 2.697±.418 | 0.762±.157 | 1.824±.144 |
| | | 128 | 0.100±..002 | **0.087**±**.019** | 2.535±.372 | 3.125±.106 | 5.552±.528 | 1.344±.131 | 1.822±.271 |
| | | 256 | **0.083**±**..011** | 0.126±.024 | 5.052±.831 | 3.768±.998 | 5.572±.584 | 4.735±.729 | 2.533±.467 |
| | Correlational (Lower Better) | 64 | **0.534**±**.110** | 0.672±.035 | 3.668±.106 | 1.653±.208 | 6.847±.083 | 1.281±.218 | 3.319±.062 |
| | | 128 | 0.521±.201 | **0.451**±**.079** | 4.790±.116 | 1.820±.329 | 6.663±.112 | 1.376±.201 | 3.713±.055 |
| | | 256 | 0.391±.146 | **0.361**±**.092** | 4.487±.214 | 1.279±.114 | 5.690±.102 | 1.800±.138 | 3.739±.089 |

blood-oxygen-level-dependent time series, Sines [5] (Yoon et al., 2019b) generated from different frequencies, amplitudes, and phases, and Mujoco [6] from multivariate physics simulation.

These datasets offer a comprehensive range of time series characteristics, including periodic and aperiodic patterns, varying dimensionality, and different levels of feature correlation, allowing for a thorough evaluation of our method across diverse scenarios.

Following practices in time generation (Yuan and Qiao, 2024),We have 4 metrics to evaluate our method: 1) **Discriminative Score** (Yoon et al., 2019b): Measures distributional similarity between real and synthetic data. A post-hoc time series classification model (2-layer LSTM) is trained to distinguish between real and synthetic sequences. The classification error on a held-out test set is reported, with lower scores indicating higher quality synthetic data. 2) **Predictive Score** (Yoon et al., 2019b): Assesses the usefulness of synthetic data for predictive tasks. A post-hoc sequence prediction model (2-layer LSTM) is trained on synthetic data to predict next-step temporal vectors. The model is then evaluated on real data, with performance measured by mean absolute error (MAE). Lower scores indicate better preservation of predictive characteristics in synthetic data. 3) **Context-Fréchet Inception Distance (Context-FID)** (Jeha et al., 2022): Quantifies the quality of synthetic time series by computing the difference between representations that fit into the local context. This metric captures both distributional similarity and temporal dependencies. 4) **Correlational Score** (Liao et al., 2020): Evaluates the preservation of temporal dependencies by comparing cross-correlation matrices of real and synthetic data. The absolute error between these matrices is computed, with lower scores indicating better preservation of temporal structure.

## 4.2 IMPLEMENTATION DETAILS

Our FM-TS model adapts the rectified flow matching approach for time series data. The architecture is based on an encoder-decoder transformer, similar to the model in Diffusion-TS (Yuan and Qiao, 2024), but with several key enhancements: QK-RMSNorm (bla, 2024), RoPE (Su et al., 2024), Logit-Normal sampling strategy (Esser et al., 2024b), Attention register (Darcet et al., 2023; Xiao et al., 2023) and Sigmoid attention (Ramapuram et al., 2024). We set the default values of alpha to 3 and k to 0.0625 (with $k$ specifically applied in conditional generation tasks). For more details, please see Supplementary materials.

## 4.3 UNCONDITIONAL TIME SERIES GENERATION

---

[5] https://github.com/jsyoon0823/TimeGAN

[6] https://github.com/google-deepmind/dm_control

We benchmarked FM-TS against other methods for unconditional time series generation across six datasets. As shown in Table 1, FM-TS consistently outperforms other methods on most evaluation metrics. On the discriminative score, FM-TS achieves 0.005, 0.019, 0.011, 0.005, 0.053, and 0.106 on the Sines, Stocks, ETTh, MuJoCo, Energy, and fMRI datasets, respectively. In comparison, the second-best method, Diffusion-TS, achieves 0.006, 0.067, 0.061, 0.008, 0.122, and 0.167 on the same datasets. This represents a reduction in discriminative score ranging from 17% to 82%, validating FM-TS's great improvement. We attribute this superior performance to the synergy of rectified flow matching with time series-specific optimizations.

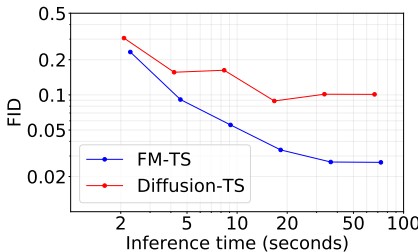

Figure 3: FID results with different $N$, the $N$ list is 1, 2, 4, 8, 16, 32.

In Table 2, we extended unconditional time series generation to longer sequences (64, 128, 256) on ETTh and Energy datasets. We observe FM-TS excels on the ETTh dataset, achieving best scores in 11 out of 12 metrics (except on Discriminative score with 256-length on Energy dataset) across all lengths, with particularly strong performance in Context-FID. On the Energy dataset, FM-TS shows mixed results, outperforming in Context-FID but a little bit falling behind Diffusion-TS in others, suggesting dataset-specific characteristics may influence its effectiveness on longer sequences.

## 4.4 EFFICIENCY BENCHMARK OF FM-TS

Compared to Diffusion-TS (Yuan and Qiao, 2024), FM-TS not only delivers superior performance across various settings but also demonstrates significantly better efficiency in both training and inference. To evaluate training efficiency, we benchmarked FM-TS and Diffusion-TS across multiple training epochs on the Energy dataset. As shown in Figure 1, We observe that FM-TS consistently achieves superior FID scores compared to Diffusion-TS, with training epochs ranging from 2,500 to 25,000. Notably, FM-TS outperforms even with as few as 30 iterations ($N = 30$), whereas Diffusion-TS can not achieve even with 200 inference steps. The observed efficiency in terms of required iterations $N$ can be attributed to the straightness property of rectified flow matching, a phenomenon extensively studied by Liu et al. (2022).

To further assess inference efficiency, we compared the final models of FM-TS and Diffusion-TS, testing different numbers of iterations ($N$) during sampling for inference. As seen in Figure 3, FM-TS not only delivers better performance but also achieves faster inference times compared to Diffusion-TS, highlighting its efficiency advantages. This empirical evidence indicates that FM-TS is capable of facilitating more rapid and accurate time series generation.

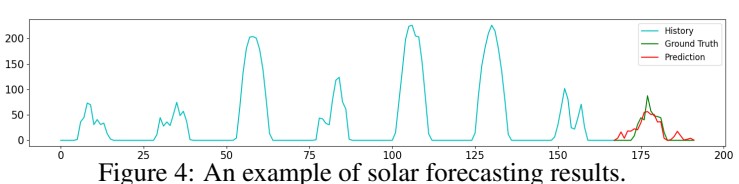

Figure 4: An example of solar forecasting results.

## 4.5 CONDITIONAL TIME SERIES GENERATION

After validating FM-TS on unconditional time series generation, we further assessed its generalizability for conditional time series generation. Instead of retraining the model, we employed the specialized inference algorithm, detailed in Algorithm 1, to incorporate observed information into inference for conditional setting. As stated in Section 3.1, conditional time series generation includes two primary tasks: forecasting and imputation. To demonstrate the effectiveness of FM-TS, following the practice in (Alcaraz and Strodthoff, 2022a) and (Tashiro et al., 2021), we benchmarked it on Solar and Mujoco datasets.

Table 3 presents the forecasting performance on the Solar dataset. Given a sequence length of 168, FM-TS achieved a superior mean-squared-error of 2.18e2 when predicting the next 24 time points, significantly outperforming the second-best model, Diffusion-TS, which scored 3.75e2. This highlights the substantial improvement in prediction accuracy and sequence alignment with FM-TS in forecasting. In Fig. 4, we presented an example of the forecasting results by FM-TS and target,

where FM-TS successfully captures the incoming peak region in the future time series. Additionally,

Table 3: Time Series Forecasting and Imputation Results

| Model | Solar Forecasting | Mujoco Imputation | |
| --- | --- | --- | --- |
| | $168 \rightarrow 24$ | Missing(70 %) | Missing(80 %) |
| GP-copula | 9.8e2 | – | – |
| TransMAF | 9.30e2 | – | – |
| TLAE | 6.8e2 | – | – |
| RNN GRU-D | – | 11.34 | 14.21 |
| ODE-RNN | – | 9.86 | 12.09 |
| NeuralCDE | – | 8.35 | 10.71 |
| Latent-ODE | – | 3.00 | 2.95 |
| NAOMI | – | 1.46 | 2.32 |
| NRTSI | – | 0.63 | 1.22 |
| CSDI | 9.0e2 | 0.24 | 0.61 |
| SSSD | 5.03e2 | 0.59 | 1.00 |
| Diffusion-TS | 3.75e2 | 0.00027 | 0.00032 |
| FM-TS | **2.13e2** | **0.00007** | **0.00014** |

we evaluated FM-TS on the imputation task (following setting of (Alcaraz and Strodthoff, 2022b)) using the MuJoCo dataset in Table 3, where it consistently outperformed other methods under varying missing data ratios. Despite most competing methods being specifically designed for conditional time series generation, FM-TS demonstrated superior performance across multiple scenarios. The Mean Squared Error (MSE) for missing rate 70% condition has decreased from 0.00027 of Diffusion-TS to 0.00007, representing a substantial 74.1% reduction.

## 4.6 Visualization comparison of FM-TS

To offer a more direct comparison between generated and target sequences, we followed the practices outlined in (Yuan and Qiao, 2024), mapping both generated and target sequences into an embedding space using PCA (Shlens, 2014) and t-SNE (Van der Maaten and Hinton, 2008). In Fig. 5a, 5d, 5b, 5e, we present a comparison of PCA and t-SNE visualizations between sequences generated by FM-TS and Diffusion-TS, as well as the corresponding target sequences. It is evident that the embeddings from FM-TS show greater consistency with the target sequences in both visualizations, highlighting the superior performance of FM-TS. We further analyzed the results using kernel density estimation (KDE) (Chen, 2017), shown in Fig. 5c and 5f. The KDE for FM-TS aligns more closely with the target sequences, especially on the right slope, where Diffusion-TS exhibits noticeable fluctuations, further validating FM-TS's superior accuracy.

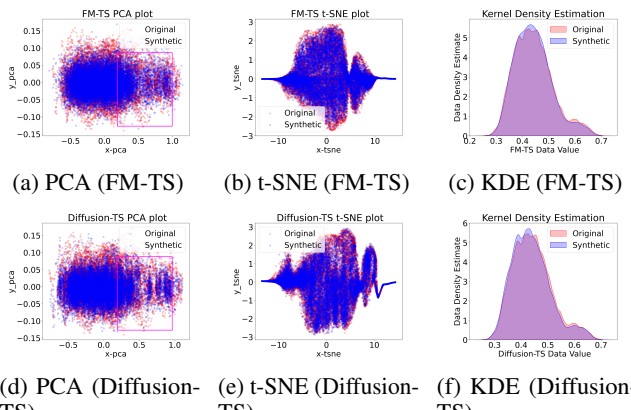

(a) PCA (FM-TS) (b) t-SNE (FM-TS) (c) KDE (FM-TS)

(d) PCA (Diffusion-TS) (e) t-SNE (Diffusion-TS) (f) KDE (Diffusion-TS)

Figure 5: Embedding visualization comparison of generated sequences by FM-TS and Diffusion-TS methods relative to the target sequences using PCA, t-SNE, and Kernel Density Estimation. Here red indicates the target sequences, where blue indicates the generated sequences.

## 4.7 Ablation Study

In this section, we will study the key components in FM-TS framework to understand their contributions.

**Logit-Normal distribution for training $t$ sampling** In Table 4, we compared the performances on energy dataset on the 4 metrics with uniform distribution and our default logit-normal distribution for

training. It is clear that logit-normal distribution shown in Fig. 2b is essential for training a stable and accurate model. That validates our assumption that distribution can encourage model to learn the hardest information.

Table 4: Training Sampling Strategy Comparison

| Method | FID | CS | DS | PS |
|---|---|---|---|---|
| FM-TS | 0.028 | 0.721 | 0.058 | 0.250 |
| uniform sampling | 0.029 | 0.676 | 0.056 | 0.250 |

**Number of iterations $N$**

The number of iteration steps is a critical factor in balancing performance and efficiency. As shown in Fig. 3, performance begins to saturate when $N = 32$. Based on a comparison with Diffusion-TS, we identify $N = 32$ as the optimal point for achieving a balance between performance and computational efficiency. That validates our assumption FM-TS is both accurate and efficient compared to that of Diffusion-TS.

$t$ **power sampling factor $k$ for conditional generation** In Algorithm 1, we proposed to use power sampling factor $k$ to control conditional time series generation. In Fig. 6, we compared different $k$ for generation under different number of inference iterations $N$. When $N$ becomes larger, which indicates the inference becomes more stable, we found that a small $k$ can lead to better performance. The effectiveness of conditional generation can be significantly improved by focusing on later sampling steps in the diffusion process. Setting $k < 1$ in $t^k$, where $t \in (0, 1]$, enables more effective conditioning. For instance, with $t = 0.25$ and $k = 0.5$, $t^k = 0.5$ represents a later time step than $t$, bringing generated samples closer to the target distribution.

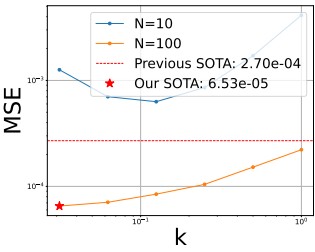 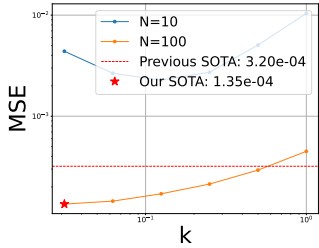 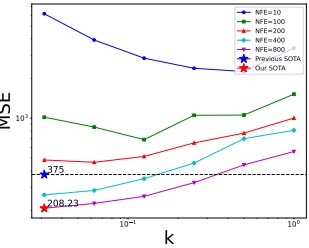

(a) MSE with changing $N$ and $k$ on solar dataset imputation tasks, with missing ratio 0.7

(b) MSE with changing $N$ and $k$ on solar dataset imputation tasks, with missing ratio 0.8

(c) MSE with changing $N$ and $k$ on Mujoco dataset forecasting tasks

Figure 6: Conditional generation with different $k$

## 5 CONCLUSION

We introduced FM-TS, a novel time series generation framework based on rectified flow matching. FM-TS achieves efficient one-pass generation while maintaining high-quality output. Experimental results demonstrate FM-TS's superior speed in training and inference, consistently outperforming state-of-the-art methods across various datasets and tasks in both conditional and unconditional generation. A key innovation of FM-TS is the novel t power sampling technique, which significantly enhances performance in conditional generation tasks. By using $t^k$ with $k < 1$, the model focuses on later steps in the generation process, allowing for more effective incorporation of conditional information. This adaptive sampling strategy proves particularly beneficial in tasks like forecasting and imputation, where FM-TS demonstrates substantial improvements over existing methods.

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
