# A   Implementation Details for FM-TS

FM-TS utilizes an encoder-decoder Transformer with the same architecture specifications as Diffusion-TS, including layer count, hidden dimension, attention heads, and feedforward dimension. It incorporates RoPE with a base frequency of 50000, an attention register with a token size of 128 for energy 256 unconditional generation, and sigmoid attention with a scaling factor of 1. The training process mirrors Diffusion-TS in terms of optimization parameters, learning rate schedule, batch sizes, and training duration. Hardware setup includes Nvidia A100x8 GPUs with CUDA 11.7. Flow Matching hyperparameters include an alpha of 3 for time shifting and a K of 0.0625 for t power sampling. Logit-Normal sampling uses a mean of 0 and standard deviation of 1. Data preprocessing involves -1 to 1 scaling, consistent with Diffusion-TS, without additional augmentation techniques.