# OpenReview forum: "FM-TS: Flow Matching for Time Series Generation"
_ICLR.cc/2025/Conference — Submitted to ICLR 2025_

### Official Review · Reviewer_dUbu · 2024-10-21

**Soundness:** 2
**Presentation:** 2
**Contribution:** 2
**Rating:** 3
**Confidence:** 3

**Summary:**

The authors incorporate Flow Matching into the diffusion model for time series modeling. They test it on multiple datasets with multiple metrics with ablation studies and efficiency tests.

**Strengths:**

1. Clear presentation.
3. Proper literature review.

**Weaknesses:**

1. The paper does not do enough efforts to distinguish itself from similar works in the literature, such as CFM-TS in ICML 2024.

2. This paper's experiment does not compare with ODE for time series works, which could also be used for time series generation. What is unique in flow matching that is beneficial for time series modeling?

3. It seems to me that the predictive score is most important -- unless the authors could suggest other usage of generated fake time series, if not for privacy-protected learning. However, the proposed model does not seem significantly better than baselines in the predictive score.

4. Figure 1 does not seem to be a comprehensive comparison in efficiency. It only compares FID against one baseline.

5. Very limited interpretation/reasoning about the experimental results. Mostly it is only about listing all the numerics, but readers can hardly understand why the result looks like the ones presented in the paper and the implication.

**Questions:**

1. What are the possible reasons that diffusion models show bar-like synthetic PCA plot in Figure 5? It is strange to have bar-like shape in a PCA plot.

2. Figure 4 does not seem to suggest good performance of FM-TS. Why is that, and why present it in the paper?

3. When you do conditional time series generation, how do you do diffusion model baseline? Is that also tuned to be conditional, or does it still learn the entire density of the time series?

---

### Official Review · Reviewer_eDwC · 2024-10-29

**Soundness:** 1
**Presentation:** 1
**Contribution:** 1
**Rating:** 3
**Confidence:** 3

**Summary:**

This paper addresses the task of generating both conditional and unconditional time series data. Diffusion models have proven effective for this purpose but are computationally expensive. To address this, the authors propose a model called FM-TS, which leverages rectified flow for efficient time series generation. A key advantage of FM-TS is its ability to generate conditional time series data without requiring retraining after being initially trained on unconditional generation tasks. The model is evaluated across multiple tasks, including unconditional generation, forecasting, and imputation, demonstrating superior performance compared to existing approaches.

**Strengths:**

**S1** Time series generation, though highly significant, remains less explored compared to image generation. The authors have made a commendable effort in addressing this challenging task.

**S2** The paper demonstrates strong experimental results in both unconditional and conditional time series generation.

**Weaknesses:**

**W1** The paper lacks clarity in presenting the core model illustrated in Figure 2. Although rectified flow is explained thoroughly as a preliminary concept, the main components of the model are only briefly introduced in the final paragraph of Section 3, without adequate explanation. The authors should explain the model components, clarify the rationale behind the chosen architecture and its relevance to time series.

**W2** In its current form, the paper appears to be an application of rectified flow to time series without addressing the specific challenges in adapting rectified flow from image data to time series such as causality of time series, seasonality, trends; limited novelty.

**W3** The experimental evaluation is insufficient.
- **W3.1** What is the motivation behind choosing squared error as the evaluation metric? Squared error is a metric of evaluation for point prediction. For a generative time series model, evaluating solely on squared error for forecasting and imputation is inadequate. A more suitable evaluation would be based on metrics like Continuous Ranked Probability Score (CRPS) for univariate or Negative Log-Likelihood for both univariate and multivariate distributions.
- **W3.2** What is the reason for choosing only the current set of baselines for forecasting and imputation? It would be beneficial to compare FM-TS against point-estimation models since the chosen evaluation metric is squared error; ex. PatchTST and TS-Mixer for forecasting tasks.
- **W3.3** For the imputation task, the choice of only 70% and 80% missing data rates should be justified. For direct comparison with existing work, consider the settings from Toshiro et al. (10%, 50%, and 90%) and Alcaraz et al. (70%, 80%, and 90%).
-**W3.4** Since the main motivation for the FM-TS is the computational inefficiency of diffusion models, authors should show the runtime of training FM-TS and other baseline models (runtime per epoch, number of epochs until convergence, and/or total training time). Figure 3 attempts to do this job in terms of inference speed only.

**Minor:**

- **M1** In line 242, should the function \( v: {R}^{l \times d} \times [0,1] \to {R}^{l \times d} \) include \( t \in [0,1] \) as an argument?
- **M2** Please increase the font size of legends and labels in Figures 4 and 5 for readability.
- **M3** Enhance the captions for Tables 3 and 4 to clarify the evaluation metrics used.

**Questions:**

Please see Weaknesses

---

### Official Review · Reviewer_Eg2z · 2024-10-30

**Soundness:** 3
**Presentation:** 2
**Contribution:** 3
**Rating:** 5
**Confidence:** 4

**Summary:**

This paper proposes FM-TS, a new approach for time series generation, based on the rectified flow. Leveraging the features of the rectified flow, FM-TS reduces the computational cost during training and handles the slow inference observed in traditional diffusion-based models. In addition, several proposed methods enable the direct use of models trained on unconditional generation for conditional tasks like forecasting and imputation without retraining. Experimental results show that FM-TS achieves better performance than existing methods in both effectiveness and efficiency.

**Strengths:**

1. FM-TS effectively utilizes rectified flow to address the high computational demands and slow inference with traditional diffusion models, offering a more efficient generation.
2. The introduction of the t-power sampling method is innovative, generalizing the generative models trained in unconditional setting to conditional scenarios without retraining.
3. Experiments for unconditional generation are well-designed and the results are solid across various metrics.

**Weaknesses:**

1. The overall writing quality is good, but some statements are confusing or misleading. For example, the descriptions about the capability of diffusion models in handling long-term dependency are contradictory in line 058 and line 061. The former states that diffusion models can capture long-range dependencies and generate diverse, high-quality samples, while the latter asserts that diffusion models struggle to preserve long-range dependencies and intricate patterns in time series data.
2. There is insufficient discussion on the conditional generation, particularly on why the unconditional models are adapted for conditional tasks by Algorithm 1. The introduction of concepts like t-power sampling lacks sufficient context and explanation, which makes it challenging for readers unfamiliar with them to understand their implication. Can the authors provide a brief example of how Algorithm 1 adapts unconditional models for conditional tasks? Can you also expand the intuition behind t-power sampling and its role in this adaptation?
3. The ablation study on the logit-normal distribution does not convincingly demonstrate its superiority; its performance is comparable or inferior to uniform sampling. Could the authors provide more analysis on why the logit-normal distribution is beneficial despite the results shown in Table 4? Are there any qualitative differences or theoretical advantages not captured by the metrics used?

**Questions:**

1. Given that the performance of the logit-normal distribution appears comparable or inferior to uniform sampling in the ablation study, can you clarify its advantages?
2. What’s the underlying mechanism that t-power sampling enables the direct application of unconditional models for conditional tasks? Are there any trade-offs?
3. What’s the runtime of the training phase and inference phase of FM-TS? How does this efficiency compare to other generative approaches, such as GANs and traditional diffusion-based models?
4. What are the primary limitations of the FM-TS model?

---

### Official Review · Reviewer_KHxJ · 2024-11-04

**Soundness:** 2
**Presentation:** 2
**Contribution:** 1
**Rating:** 1
**Confidence:** 5

**Summary:**

This paper proposes FM-TS, a framework for time series generation based on Flow Matching (FM), as an alternative to diffusion models. The authors argue that FM-TS addresses the computational inefficiency and complexity of diffusion models by simplifying the generation process through continuous trajectory optimization. FM-TS is presented as being able to support both conditional and unconditional time series generation without retraining. However, significant gaps in the paper’s theoretical foundation, experimental validation, and clarity raise questions about the viability and originality of the approach.

**Strengths:**

- The paper makes an interesting attempt to apply Flow Matching, a technique that has shown promise in image generation, to the complex domain of time series generation.

- The paper claims substantial efficiency gains over diffusion-based methods. Diffusion models, while powerful, suffer from high computational costs due to their iterative nature. Flow Matching theoretically offers a more straightforward ODE-based path, which could reduce the number of forward passes required for inference and training.

**Weaknesses:**

- The paper’s flow and organization present significant challenges in readability, largely due to unclear transitions between key concepts such as computational efficiency and generalization. It seems that the authors do not consistently differentiate these concepts in their approach. The dense and complex sections lack clear explanations, detracting from the paper’s overall coherence.

- There is an unusual conflation of generalization and computational requirements, resulting in ambiguity. For instance, the authors assert that diffusion contributes to generalization on lines 056–057, yet they appear to refute this on lines 057–058, leading to further confusion.

- The paper also lacks reproducibility experimentation (**no available code or scripts**), as no implementation details or code are provided. Including code would facilitate verification of the results and support a broader understanding. Moreover, the appendix consists of only a few lines of explanation in general. It seems that the paper is not ready for publication at this stage.

- The claim that FM-TS can generalize to conditional generation tasks without retraining is intriguing but underdeveloped. The paper lacks comparisons with models specifically designed for conditional tasks, and no compelling evidence is presented to validate FM-TS’s performance in such scenarios. A deeper exploration of FM-TS’s generalization capability would strengthen this claim.

- The authors assert that their model outperforms the current state-of-the-art (SOTA); however, the results in Table 2 do not support this claim, as high standard deviation values suggest inconsistent performance. It would be valuable for the authors to discuss these variations and integrate them into their analysis.

- The paper suggests that imputation and forecasting tasks are nearly identical, differing only in the choice of point masking $M$. This assumption oversimplifies the nature of imputation, which often requires bidirectional information to accurately infer missing points. In contrast, forecasting typically operates with unidirectional data. Recognizing these differences is essential for model design and performance.

- The implementation details of "t power sampling" are missing. Without an explanation of how this method improves results, it is difficult to assess its functional role. Providing a detailed description of the sampling process would enhance transparency and reproducibility, offering insight into whether this is an optimization layer or a refinement in sampling for conditionality.

- The paper concludes with a vague claim that the unconditional model can be “directly used for conditional generation.” However, no details or references are given to substantiate how the model adapts to conditional tasks without retraining. A brief explanation or citation would clarify this point.

At this stage, it is challenging to recommend acceptance of this paper, primarily due to concerns regarding reproducibility. Without access to code, it remains unclear how to replicate the authors' results. Furthermore, ***the improvements in the paper's tables do not align well with the contributions claimed in the introduction.***

**References**

[1] Qi, M., Qin, J., Wu, Y., & Yang, Y. (2020). Imitative non-autoregressive modeling for trajectory forecasting and imputation. In Proceedings of the IEEE/CVF Conference on Computer Vision and Pattern Recognition (pp. 12736-12745).

**Questions:**

- Could you clarify how the drift function $ v(Z_t, t) $ is modeled, and why you chose a linear interpolation $ Z_t = t \cdot Z_1 + (1 - t) \cdot Z_0 $? How does this linear interpolation impact the model’s ability to capture complex time dependencies in non-linear time series data?

- The authors claim that this work represents a novel contribution to the field of Flow Matching. However, how does it build on or differ from the existing work presented in [2]?

- The authors assert that the unconditional model can be directly applied to conditional generation without retraining. Could you elaborate on the mechanisms or transformations that enable this adaptation? Does this adaptation require additional architectural components, or is conditional information handled implicitly by the model?

**References**

[2] Kerrigan, G., Migliorini, G., & Smyth, P. (2023). Functional flow matching. arXiv preprint arXiv:2305.17209.

---

### Meta-Review · Area_Chair_W1NU · 2024-12-19

**Metareview:**

This paper introduces a new generative framework for time series generation based on Flow Matching. The approach is demonstrating potential advantages over diffusion models, and other recently introduced generative models for time series. Interestingly, the authors provide results for both unconditional and conditional generation scenarios. While the approach itself is of interest, all reviewers identified several weaknesses, including technical and organizational issues. Some statements were found to be confusing or misleading, and concerns were raised regarding the quality of the experimental evaluation. Unfortunately, the authors have not addressed these points.

Given these shortcomings, as reflected in the low review scores, I recommend rejecting this submission.

**Additional Comments On Reviewer Discussion:**

No changes were made during the rebuttal period.

---

### Decision · Program_Chairs · 2025-01-22

Reject